

# Above-treeline ecosystems facing drought: lessons from the European 2022 summer heatwave

Philippe Choler[1]

[1] Univ. Grenoble Alpes, Univ. Savoie Mont Blanc, CNRS, LECA, F-38000 Grenoble, France

*Correspondence to*: Philippe Choler (philippe.choler@univ-grenoble-alpes.fr)

**Abstract.** In 2022, Europe experienced an extremely dry and hot summer. In the Alps, this episode occurred after an unusually low snowfall winter, which aggravated the dryness of soils. This study examines the impact of this particular year on the canopy greenness of above-treeline ecosystems by comparison with previous heat waves that hit the Alps during the last two decades. Normalized Difference Vegetation Index (NDVI) time series derived from the MODIS satellite were processed to

extract the temporal variability of yearly maximum NDVI (NDVImax). The responsiveness of NDVImax to snow cover duration and growing season weather conditions was evaluated in contrasting hydro-climate regions of the Alps using linear mixed effect models. The year 2022 was unique in that the summer heat wave led to a widespread negative anomaly of NDVImax. The magnitude of this anomaly was unprecedented in the southwestern, driest part of the Alps, where vegetation activity was found to be particularly responsive to snow cover duration and early summer precipitation. In the colder and

wetter regions, all warm to very warm summers before 2022 had led to increased canopy greenness, but the combination of a reduced snow cover and low early summer precipitation counteracted this expected beneficial effect in 2022. This study provides evidence that the control of canopy greenness by temperature and water balance differs markedly across regions of the Alps and that the year 2022 bears witness to a shift toward an increasing importance of moisture availability for regulating plant growth at high elevation. This is viewed as a warning sign of what could become the new norm in the years ahead in the

context of increasing frequency and intensity of extreme droughts throughout temperate mountain ecosystems.

## 1 Introduction

A severe heat wave and drought hit a large part of the Northern Hemisphere during the 2022 summer (Lu et al., 2023). Over the last two decades, similar long-lasting warm and dry summer events have been recorded in Europe, such as in 2003 and 2015, and the recurrence of these events has no equivalent in the last centuries (Buntgen et al., 2021). In a warmer climate, it

is widely acknowledged that the duration, frequency and intensity of extreme meteorological events will increase and that an increasing proportion of lands will be affected (Coumou and Rahmstorf, 2012; Russo et al., 2014). The combination of drought and heat has an overall negative effects on the gross primary productivity of terrestrial ecosystems (Von Buttlar et al., 2018). However, a range of factors are known to modulate ecosystem responses, for example the type of vegetation, its sensitivity to water and temperature, the phenological period during which extreme events occurr and the soil moisture content during warm



episodes (Sippel et al., 2018; Sippel et al., 2016; Von Buttlar et al., 2018). Most of these factors operate at a local scale, rendering a comprehensive analysis of these impacts particularly challenging.

The primary productivity of above-treeline ecosystems is primarily controlled by temperature and length of the growing season (Churkina and Running, 1998; Choler, 2015; Myers-Smith et al., 2015). Over the last decades, most of these ecosystems have benefitted from warmer conditions during the snow free period, and this all the more so as the rate of warming is particularly

pronounced at high elevation (Beniston, 2005; Pepin et al., 2015). As a result, a long-term increase of fractional vegetation cover and primary productivity has been documented in temperate mountains using remote sensing studies (Choler et al., 2021; Anderson et al., 2020; Zhong et al., 2019) and vegetation surveys (Rogora et al., 2018; Steinbauer et al., 2018). In terms of seasonal variation, several reports have outlined the positive effects of very warm summers on the peak productivity of high elevation ecosystems (Jolly et al., 2005b; Corona-Lozada et al., 2019). However, these reports also highlighted marked regional

differences in ecosystem responsiveness, suggesting that other factors in addition to temperature are modulating the interannual variations of their primary productivity. When drought coincides with heat wave, the positive effects of a warm summer fades away, and this is more likely to happen in the driest and warmest parts of mountain range (Jolly et al., 2005b; Corona-Lozada et al., 2019; De Boeck et al., 2016; Cremonese et al., 2017). Other reports examining ecosystem phenology (Fu et al., 2015) or radial growth of mountain shrubs (Francon et al., 2021) also suggested that carbon gain has become more sensitive to water

availability in the last decades. All these results call for further investigation to determine where and when water availability becomes a co-dominant limiting factor of primary productivity at high elevation.

Above-treeline temperate ecosystems are seasonally-snow covered ecosystems, characterized by complex interactions between snowmelt dynamics and vegetation properties, including plant distribution, growth and functional traits (Jonas et al., 2008; Walker et al., 1993; Choler, 2015). On the one hand, early snowmelt can be beneficial to canopy greenness as it increases the

favorable period for carbon gain. On the other hand, it may exacerbate the dryness of soils and the exposure to early frosts, both of which are detrimental to plant growth (Francon et al., 2020). Regarding prolonged snow cover duration, the negative effect of a short growing season can be partially or totally compensated by an increasing rate of early growth because of enhanced water supply and nutrient availability liberated by the snowpack during the optimal warm growth period. Understanding this interplay between snow cover duration and meteorological conditions during the early summer is pivotal

in order to disentangle the underlying causes of variability in the vegetation productivity of above-treeline ecosystems.

Long-term ground surveys of primary productivity are notoriously difficult to carry out in complex high elevation terrain and the few data available cannot inform on the trends and their geographical variations at the massif scale. Earth observation is the only way to evaluate how ecosystems have responded to extreme events and to what extent this response varies across regions. Medium resolution remotely-sensed data with daily revisiting times have proven useful to track snow cover dynamics

and vegetation activity in arctic and alpine ecosystems (Beck et al., 2006; Choler, 2015; Xie et al., 2020). Specifically, the peak value of vegetation indices is commonly used as a proxy of annual maximum canopy greenness, capturing vegetation growth occurring during the first part of the growing season (Tucker, 1979; (Rossini et al., 2012). This approach provides an



avenue for developing empirical models of canopy greenness with the aim of unraveling the relative importance of snow and meteorological drivers on vegetation activity at broad spatial scale (Choler, 2015; Xie et al., 2020; Fu et al., 2021).

In the European Alps, previous remote sensing approaches have mainly focused on the long-term trends of greenness (Choler et al., 2021) and between-site variations in ecosystem phenology (Xie et al., 2020). By contrast, there has been no comprehensive study examining the interannual variability of canopy greenness, its sensitivity to extreme events and the extent to which this sensitivity differs among regions. The European Alps provide an interesting case study to address these questions given the variety of hydro-climate sub-regions (Rubel et al., 2017). The Alps exhibit strong gradients of precipitation between

the wet external ranges and the rather dry inner alpine valleys (Isotta et al., 2014), and also the influence of the Mediterranean climate in southern regions whereas the north eastern ranges are exposed to oceanic influences (Brunetti et al., 2006; Hiebl et al., 2009). In this context, I addressed three main questions: (i) what were the effects of the 2022 heat wave and drought on the canopy greenness of above-treeline ecosystems in the European Alps and to what extent did these effects differ from those observed during previous extreme events? (ii) What is the sensitivity of these ecosystems to interannual variation in snow

cover, water availability and temperature? And (iii) does the relative strength of these drivers differ between the contrasting bioclimatic subregions of the European Alps?

    I delineated bioclimate clusters for the European Alps using growing season climate variables. Then, I evaluated the variability of canopy greenness using the annual peak value of the Normalized Difference Vegetation Index (NDVImax) of the Moderate Resolution Imaging Spectroradiometer (MODIS) for the period 2000-2022. I used the ERA5-land re-analyses to evaluate early

summer meteorological conditions. Finally, I developed a linear mixed effect (lme) model to assess the effects of snow cover duration, water stress and temperature on the variability of productivity in the different bioclimate clusters and over the MODIS period.

## 2. Material and methods

### 2.1 Study area

This study focuses on the above-treeline ecosystems of the European Alps, a mountain range stretching over 1,200 km from Nice (France) to Vienna (Austria). The 250 m resolution MOD09Q1 products in the native sinusoidal projection was used as a reference grid to select the pixels of interest based on ancillary data. I used the Tree Cover Density of year 2018 (https://land.copernicus.eu/pan-european/high-resolution-layers/forests/tree-cover-density/) to extract non-forested pixels (<5%), and a 25 m resolution Digital Elevation Model to discard non-forested pixels below the natural treeline. The elevation

of the natural treeline was set for each latitudinal band according to the polynomial regression given in (Korner, 1998). To account for locally lower treeline positions, I also included non-forested pixels located 100 m lower than the regression line. Very sparsely vegetated pixels with a long-term average NDVImax below 0.15 showed inconsistent annual peaks of greenness and were discarded. Densely vegetated pixels with a NDVImax above 0.75 were also removed because the sensitivity of NDVImax to canopy greenness decreases when biomass is high, due to saturation issues (Huete et al., 2002). The final dataset





comprised 227,318 pixels. According to the 100 m resolution land cover product of Copernicus Global Land Service (CGLS) (Buchhorn et al., 2020), the selected pixels were a majority of herbaceous vegetation and a minority of sparsely vegetated areas (Supplement Table 1). I calculated pairwise dissimilarities between NDVI time series using Euclidean distance and used the Partition Around Medoid algorithm of the *Cluster* R package (Maechler et al., 2022) to identify clusters. This allowed to check that selected pixels exhibited the characteristic phenology of seasonally-now covered vegetated ecosystems, i.e. with an

abrupt rise of greenness after snow melt and a maximum NDVImax achieved at the end of July – early August (Supplement Fig. S1).

### 2.2 Snow and climate datasets

I used CHELSA v2.1 and CHELSA-BIOCLIM+ datasets (Brun et al., 2022; Karger et al., 2021) to retrieve the climate averages of the reference period 1981-2010. CHELSA is a high resolution climatology that provides downscaled surface variable

estimates at a horizontal resolution of 30 arc sec (Karger et al., 2017). The following variables were used to characterize the summer (June, July and August) climate in the study area: mean daily air temperatures (bio10), mean monthly precipitation amount (bio18), surface downwelling shortwave radiation (rsds) and vapour pressure deficit (vpd). To identify climate subregions, I computed pairwise dissimilarities between observations using Euclidean distance and performed a cluster analysis with the Partition Around Medoid algorithm of the *Cluster* R package (Maechler et al., 2022). I varied the number of

prescribed clusters from 4 to 9. For the sake of simplicity, results are only presented for 6 clusters. This was a fair compromise between a fine scale regionalization and a sufficient number of pixels per cluster for further analyses.

I used ERA5-Land gridded datasets to estimate early summer meteorological conditions during the MODIS period. ERA5-land is a process-based, climate re-analysis providing 0.1° resolution variables related to the energy and water balance of land surfaces (Muñoz-Sabater et al., 2021). The study area encompassed 1053 ERA5-land cells (Supplement Table 1). I extracted

the pre-calculated monthly means of air temperature (T2M), monthly accumulation of precipitation (PRE) and potential evapotranspiration (PET) for the months of June and July, i.e. the early growing season. In the absence of available data on soil water capacity, I calculated the difference between PRE and PET, which represents atmospheric water balance used as a proxy of plant water availability during the early summer, and which is hereafter referred to as WBA. Using daily products, I also computed a Heat Wave Index (HWI) for the June-July period. This index, proposed by (Russo et al., 2014), accounts for

the magnitude and duration of heat waves. It was calculated following the simplified methodology described in (Corona-Lozada et al., 2019). Briefly, heat waves corresponded to sequences of at least three consecutive days during which the maximum daily air temperature was above the 8th decile for the reference period (1981–2010). For each heat wave, I computed the difference between the daily maximum air temperature and the 8th decile and summed all these differences for the June-July period.

To evaluate the interannual variability of snow cover extent, I extracted the number of snow-covered days during the first six months of the year from the 8-day composite MODIS Terra (MOD10A1) collection 6 products (Riggs et al., 2016). The



estimate is based on a 500-m resolution Normalized Difference Snow Index, an indicator of the snow cover that uses the blue and middle-infrared bands (Salomonson and Appel, 2004).

## 2.3 Variability of MODIS-derived NDVImax

The study exploits MOD09Q1/Terra collection 6 products, which consist of 250 m resolution 8-day composite of the Moderate Resolution Spectro Radiometer (MODIS) sensor onboard the Terra satellite. This product has proven relevant to capture abrupt greenness changes during the short growing season above the treeline (Choler, 2015). Data are distributed by the Land Processes Distributed Active Archive Center (https://e4ftl01.cr.usgs.gov/). I downloaded images from 18 February 2000 to 31 December 2022 for the tiles h18.v4 and h19v04 to cover the European Alps. Reflectance values ($\rho$) in the red and infrared

bands that were produced with high quality (according to the MOD09Q1 Quality Control flag) were used to generate time series of raw NDVI, i.e. the ratio ($\rho_{NIR} - \rho_{RED}$)/($\rho_{NIR} + \rho_{RED}$). A vast range of techniques now exist to reduce the noise of these NDVI time series and it is important to examine the sensitivity of the retrieved NDVI metrics to the chosen algorithms (Zeng et al., 2020). Here, I compared two curve fitting methods: the Savitzky-Golay (SG) smoother (Savitzky and Golay, 1964; Chen et al., 2004) based on a local window, and the Whittaker (Wh) smoother (Eilers, 2003; Whittaker, 1922) that

utilizes a penalized least square regression on the whole time series. For the SG smoother, I varied the windows size from 7 to 14 observations and a polynomial degree (filter order) of n=3. For the Wh smoother, I varied the lambda parameter from 5 to 15 and used a second order difference. The weight update function of TIMESTAT was used for successive iterations (Jonsson and Eklundh, 2004) and I compared results after 3 and 5 iterations. Finally, I extracted the following annual metrics from the denoised time series and for each pixel: the start of the season (SOS) as the first date of the year when NDVI surpasses 15%

of NDVImax and the maximum NDVI of the growing season (NDVImax). I used the *MODIStsp* R package (Busetto and Ranghetti, 2016) to download the native hdf MOD09Q1 files and the *Phenofit* R package (Kong et al., 2023) to process raw NDVI times series.

As this study focuses on variability, an important initial step consisted of detrending NDVImax time series. The NDVImax variability for a given data span is defined as the residue of the NDVImax after the removal of the trend (Wu et al., 2007). For

each pixel, I fitted a monotonic function of NDVImax over time using both the whole data span (22 years) and subperiods of 13 years. I used either the non parametric Theil-Sen estimator or a least-squares estimator to assess the linear trend, the first being often preferred when data strongly depart from normality (Hirsch et al., 1991). The final results were not affected by this choice, and I only showed the results produced with the non parametric method implemented in the *Kendall* R package (Mcleod, 2005). Our previous study based on the same raw dataset (Choler et al., 2021) showed that NDVImax trends were

robust to a +/- 5% perturbation of the red and infra-red reflectances, a value which corresponds to uncertainties associated with MODIS products (Miura et al., 2000) and so I did not reiterate this numerical simulation for this paper.

To test for a collective, or per cluster, significance of anomaly, I used a test based on the counting of signs (Huth and Dubrovsky, 2021). This test allows for fast computing and performs as well as other tests such as the False Detection Rate (Wilks, 2016) when spatial autocorrelation is moderate (Huth and Dubrovsky, 2021). To limit spatial autocorrelation, all tests



were based on the random sampling of 10% of pixels per cluster and I reported the median value of the statistics. The null hypothesis states that the number of pixels showing a positive anomaly will not significantly differ from the number of sites showing a negative one and therefore will follow a binomial distribution with parameters p = 0.5 and N the number of trials corresponding to the number of sites that were randomly sampled per cluster. A two-tailed binomial test allowed for calculating the probability of the alternative hypothesis, i.e. that positive anomalies may be either greater than or less than negative

anomalies.

**2.4 Linear mixed effect modelling**

I implemented a linear mixed effect (lme) model to characterize the variability of NDVImax over time within pixels and its variation between clusters. Lme models extend linear models by allowing for fixed and random effects in a hierarchical design. The lme model can be expressed as following:

$Y = X\beta + Z\gamma + \varepsilon$      (1)

where Y is the response, or dependent variable, X are predictors, or fixed effects, $\beta$ are the size of the fixed effects, Z is a matrix of random effects that depends on data structure, $\gamma$ is the size of the random effects, and $\varepsilon$ is a vector of unobserved random errors. I tested the following fixed effects: snow cover (MSE), air temperature (T2M), heat wave index (HWI), precipitation (PRE) and atmospheric water balance (PRE-PET). I also included an interaction term between a temperature-

related variable and a water-related variable. Random effects included clusters as the main effect, hereafter CLU, and an interaction between cluster and cells specifying that cells are grouped within clusters. Because NDVImax and predictors were not available at the same spatial resolution (250m vs. 9 km for ERA5, and 500m for MSE), I computed the median value of NDVImax per ERA5 cell and used this median as the response variable. When ERA5 cells included MODIS pixels belonging to different bioclimate clusters, I assigned to that cell the cluster containing the highest number of pixels. Complementary lme

models were also fitted for specific subsets of MODIS pixels based on (i) the long-term average of NDVImax (0.15-0.35; 0.35-0.55; 0.55-0.75) to enable comparison of ecosystem responses along a gradient of fractional vegetation cover, and (ii) the value of the Diurnal Anisotropic Index (DAH) to enable comparison of ecosystem responses along a gradient of exposure to solar radiation (Böhner and Antonić, 2009).

All models were fitted using standardized anomalies of the response and the predictors. A stepwise procedure was used in

which I first tested whether a random structure was justified and subsequently, whether including a particular fixed effect or an interaction between fixed effects was justified. The comparison of competing models was based on the Akaïke Information Criteria (AIC) and I retained models with the lowest AIC score. Models were fitted by maximizing the restricted log-likelihood with the *lme* function of the *nlme* R package (Pinheiro et al., 2022). The Broyden–Fletcher–Goldfarb–Shanno (BFGS) algorithm was chosen for nonlinear optimization. Model parameters, variance explained by the fixed effects, or marginal

variance, and variance explained by both fixed and random effects, or conditional variance, were estimated using the *MuMIn* R package (Barton, 2023).





## 3 Results

The cluster analysis of summer climate revealed a marked precipitation gradient that stretched from the southwestern to the
northeastern Alps (Fig. 1). Along with the decrease of precipitation, there was a strong increase of short-wave radiation. These
two gradients were not only driven by latitude. For a given latitude, external ranges are wetter and less sunny than the dry
inner valleys, indicating that the well-known rain shadow effect was correctly captured by the Chelsa high-resolution climate
model. The summer temperature gradient did not fully coincide with the precipitation gradient because of an overlapping effect
of elevation occurring locally. This explained for example the difference between the two dry clusters (2 and 3), the latter
including pixels at lower elevation compared to the former. In the colder and wetter context of the North-Eastern Alps, the
same contrast was observed between clusters 5 and 6 (Fig. 1).

NDVImax anomalies for the period 2000-2022 are presented for each climate cluster (Fig. 2). The year 2022 was characterized
by a widespread, negative NDVImax anomaly and by the lowest NDVImax anomaly ever recorded for cluster 1. This was in
sharp contrast with the 2003 and 2015 summer heat waves where the NDVImax anomalies were either significantly positive
(clusters 3,4,5,6) or non-significantly different from 0 (cluster 1, 2). Noticeably, the ranking of NDVImax anomalies during
these three warm years closely aligned with the warm/dry to cold/wet gradient, i.e. there was an increasing value of NDVImax
anomaly as one goes from cluster 1 to cluster 5(or 6). An opposite pattern was found for cold and wet years, such as in 2013,
2014 and 2021, where the NDVImax anomaly of dry and warm clusters was systematically above that of cold and wet ones
(Fig. 2).

Figure 3 presents the relationships between anomalies of NDVImax and anomalies of snow cover duration and early summer
conditions for clusters 1 and 6, i.e. the two clusters lying at the extremes of the climate gradients (see Supplement Fig. S3 for
the other 4 clusters). For cluster 1, the four best years for NDVImax were associated with a positive water balance (Fig. 3b).
The year 2011 was an exception, which could be explained by the relatively cold conditions prevailing in the early summer
(Fig. 3c). The two worst years for NDVImax were 2022, which combined early snow melt with a negative water balance and
high temperatures later in the summer, and 2001 which was characterized by very early snowmelt (Fig. 3a). For cluster 6, the
opposite situation was found with three of the four warmer summers (2003, 2006, 2010 and 2015) associated with a
significantly positive NDVImax anomaly (Fig. 3f). In 2006, it is plausible that this effect was offset by a very negative water
balance (Fig. 3e). As for cluster 1, early snowmelt was associated with a negative anomaly, although the year 2006 marked an
exception (Fig. 3d).

Results from the linear mixed-effect model provided a quantitative analysis of the drivers of NDVImax variability. Models
including a hierarchical random structure, i.e. clusters and cells within clusters, performed significantly better than models
without (Table 1). The water-related variable (WBA), and the temperature-related variable (T2M) were better predictors than
PRE and HWI, respectively (Table 1). The model exhibiting the lowest AIC included the three fixed effects (MSE, WBA and
T2M), the interaction between WBA and T2M, and random slopes. The variance explained by the fixed effects, or marginal
variance, was 9.1 % and that including fixed and random effects, or conditional variance, was 14.7 % (Table 1). Overall, the





fixed effects MSE, WBA and T2M were significantly positive and the interaction between WBA and T2M was significantly negative (Fig. 4a, Supplement Table 2). These findings were highly robust to the parameters used for NDVI curve fitting methods (Supplement Table 3).

Noticeably, the size of the random effect cluster showed considerable variations for WBA and T2M (Fig. 4b). For WBA, there
was a shift from high to low sensitivity along the southwest to northeast gradient (clusters 1 to 6), and an inversely related shift from low to high sensitivity for T2M. The negative interaction between WBA and T2M was stronger in the southernmost clusters 1 and 2, meaning that the negative effect of temperature was amplified when the positive effect of water availability was strong. By contrast, a positive interaction for cluster 5 was indicative of a synergistic effect of temperature and water availability. The lowest sensitivity to MSE was detected in cluster 5, which combines cold and wet conditions (Fig. 1).
The comparison of lme models fitted for the first (2000-2012) and the second (2011-2022) periods showed a slight increase of the fixed effect estimates (Fig. 4c). Changes were more noticeable changes at the cluster level, with an increasing sensitivity to MSE and WBA for the two driest clusters 1 and 2 and a somewhat reverse pattern for the other clusters (Fig.4d). Finally, temperature showed more positive effects in the recent period for clusters 5 and 6, whereas trends were negligible for the other clusters.

**4 Discussion**

Using a remote sensing approach, I carried out a comprehensive analysis of year-to-year variation in canopy greenness of above-treeline ecosystems in the European Alps, with a special emphasis on vegetation responses to the summer heat waves and extreme droughts that have affected the range during the last two decades. First, I showed that the positive effect of warm summers on plant growth vanishes when water becomes limited. This was particularly noticeable in 2022, a year of negative
NDVImax anomaly that sharply contrasted to what happened during the previous heat waves in 2003 and 2015. Second, the shift from water-limited growth in the southwestern Alps to temperature-limited growth in the northeastern Alps parallels the shift from warm/dry to cold/wet bioclimate gradient in the range. Finally, I provided evidence that water-limited ecosystems are increasingly sensitive to interannual variations in water availability whereas the most temperature-limited ecosystems are still benefitting from recent warm summers.
Phenomenologically, between-site differences in NDVImax anomalies have two possible causes: (i) contrasting exposure to the key factors controlling canopy greenness and (ii) different sensitivity of ecosystems to these factors. The lme model supported the second explanation, considering that the fixed effects, which quantify the sensitivity of canopy greenness to drivers, differed between the bioclimate clusters. Global scale assessments of vegetation sensitivity to climate have pointed out the high sensitivity of arctic and alpine ecosystems to temperature and cloud cover (Higgins et al.; Seddon et al., 2016).
The present study provides a more balanced picture, accounting for water availability and highlighting important sub-regional variability in the drivers/limiting factors of canopy greenness for above-treeline ecosystems. The spatial variability was consistent with Liebig's law of the minimum, which states that growth is primarily controlled by the most limiting resource.





In the southernmost part of the study area, water balance is a strong co-limiting factor of canopy greenness and the high temperatures recorded during heat waves exacerbate the negative impact of drought on NDVImax. In the northeastern part of

the massif, vegetation activity still benefits from hot summers, a finding that aligns with previous observations of cold ecosystems (Jolly et al., 2005b).

Although the model captured broad-scale patterns of ecosystem responsiveness, a large part of the NDVImax variability remained unexplained, which could be attributed to multiple causes. First, land surface meteorological variables are notoriously difficult to model at high elevation because of the scarcity of available observations and the importance of processes that

depend on topography and that are not accounted for by climate re-analyses (Vionnet et al., 2019). The ERA5-land products have proven useful for capturing the main features of surface variable trends in the European Alps (Monteiro and Morin, 2023), and have outcompeted other climate gridded products, such as E-OBS, in areas where the density of weather stations is low (Bandhauer et al., 2022), as is the case in the southwestern Alps. However, its coarse spatial resolution limits its usefulness for representing local heterogeneity in bioclimatic conditions on the ground. Further efforts should strive to reduce the mismatch

between the spatial resolution of climate drivers and that of remotely-sensed vegetation indices. One possible approach includes the statistical downscaling of climate variables to MODIS resolution using a digital elevation model (Baba et al., 2018).

Further unexplained NDVImax variability could be attributed to the fact that each pixel includes a variety of plant communities that may respond differently to climate. To tackle this issue, we need a deeper and more process-based understanding of

interactions between vegetation and climate variables, in order to produce microclimate layers at the landscape scale (Zellweger et al., 2019; Lenoir, 2020). Several studies have emphasized how biotic-abiotic interactions shape the microclimate variables that are key for controlling the functioning of cold ecosystems, such woody vegetation trapping windblown snow (Beringer et al., 2001; Sturm et al., 2001; Lett et al., 2020). Of utmost importance will be improved modelling of soil moisture and soil temperature dynamics in heterogeneous mountain landscapes. Unfortunately, this is currently hampered by our very

poor knowledge of key mountain soil properties, such as soil water capacity or thermal conductivity. Finally, we overlooked the possibility that variability in canopy greenness might reflect interannual variability in land management practices. Summer grazing is the dominant form of land-use in the study area. Although a decline of mountain livestock systems has been reported in parts of the European Alps (Tappeiner et al., 2008), no dataset is available to depict trends and anomalies in stocking rates and land management at the scale of the European Alps. So far, there has been limited evidence that grazing activity has a

significant effect on the NDVImax of mountain grasslands in the southwestern Alps (Carlson et al., 2017), probably because extensive grazing by sheep predominate. Further studies should examine whether similar patterns are observed in more intensively managed livestock systems.

My findings pointed out an overall positive effect of snow cover on canopy greenness. Snow cover is a complex ecological factor having multiple direct and indirect effects on vegetation activity and phenology (Gao et al., 2013; Choler, 2015; Walker

et al., 1993). Control of the length of the growing season, reduced risk of early frosts because of snow's insulating properties, and supply of water and nutrients by the melting snowpack are among the most commonly cited processes. While many studies





have examined the sensitivity of the timing of spring leaf unfolding to snow cover duration (Jolly et al., 2005a; Fu et al., 2021; Fu et al., 2015; Currier and Sala, 2022; Xie et al., 2020; Stockli and Vidale, 2004), the effect of snow cover duration on the maximum canopy greenness, i.e. peak biomass, has received less attention. Trujillo *et al.* (2012) reported on the positive

response of mountain forests to snowy winters in California's Sierra Nevada. Similar responses were recorded in the forested ecosystems of central Siberia (Grippa et al., 2005). It was also suggested that negative anomalies of snow cover do not cascade to increased canopy greenness likely because the intrinsic growth constraints limit the ability of alpine plants to benefit from early snowmelt (Baptist et al., 2010). In mountain grasslands, I previously showed that a low amount of NDVImax variance was explainable by the interannual variations of snow cover duration, essentially because faster growth after a delayed

snowmelt and a slower growth after an early snowmelt both resulted in similar NDVImax values. However, this study was carried out on a more limited spatial (French Alps) and temporal (2000-2012) scale (Choler, 2015). The present results suggest greater sensitivity to snow cover, especially in the driest areas. It is likely that negative anomalies of snow cover duration in the recent warm years resulted in more detrimental effects than before, because of increasing evaporative demand during the early summer. In the context of declining snow cover in the European Alps (Matiu et al., 2021), one may expect increasing

detrimental effects of early snowmelt, particularly in the driest parts of the range.

Over the last two decades, a significant increase of NDVImax has been detected for most of the above-treeline ecosystems of the European Alps (Choler et al., 2021). Similar greening trends have been detected in arctic ecosystems though with large spatial heterogeneity (Berner et al., 2020; Myers-Smith et al., 2011). In the Alps, these trends were attributed to the combination of summer warming, maintained snow cover duration at high elevation (above 1800 m) and the density-dependence nature of

plant cover increase. Interestingly, the southwestern Alps, which are shown here to be the most sensitive region to drought, were also previously identified as a hotspot of greening (Choler et al., 2021). The present results suggest that recurrent dry years may halt or even reverse this greening trend, calling for the need for ongoing earth observation monitoring in order to identify potential break points (Filippa et al., 2019). Recent observed decreases in global NPP trends have been attributed to drought in the context of water-sensitive ecosystems (Zhao and Running, 2010). D(De Jong et al., 2013) observed transitions

from greening to browning trends in several water-limited regions including Patagonia, the Sahel and northern Kazakhstan, among others. Even if no major trend in precipitation has so far been detected in the Alps over the last decades, warmer summer temperatures will lead to increased evaporative demand, with detrimental effects on vegetation activity in the driest regions. Further studies are needed to further investigate how decadal-trends in climate and short-term responses of vegetation to meteorological extremes interact to influence trajectories of high elevation plant growth. A recent account from Poppe & al.

(2023) demonstrated increasing water use efficiency in European grasslands, with the most positive trends observed in highly productive grasslands. In areas combining fast greening (i.e. trends of primary productivity) and evidence of water limitation, it would be of particular interest to examine whether vegetation changes lead to a functional shift toward drought tolerance and to changes in water use efficiency. In the European Alps, the Global Observation Research Initiative in Alpine Environments (GLORIA) provides long-term surveys of the summit flora (Pauli et al., 2005) and offers opportunities to tackle

this issue providing water-related functional plant attributes are documented.

**Conclusion**

This study shows that the drought accompanying the most recent 2022 heat wave had unprecedented negative impacts on the canopy greenness of above-treeline ecosystems in the European Alps. This main finding emerged from a broader attempt to

improve our understanding of the drivers of interannual variation of vegetation activity and their geographical variations across different sub-regions of the Alps. Although the developed model proved useful to capture the contribution of temperature and water related variables at this broad scale, it felt short of predicting the substantial amount of local variability that is detectable using remote sensing. Efforts should be devoted toward improved representation of the local conditions that matter for plant growth in complex mountain landscapes. In this line of thought, better accounting for the interplay between snowmelt

dynamics, soil water availability and local temperature is pivotal. I anticipate that the implementation of such fine-grained soil climate layers will enable us to take greater advantage of remotely sensed proxies of vegetation activity, allowing us to develop empirical models of ecosystem functioning and to forecast how these ecosystems will respond to the increasing frequency and intensity of extreme meteorological events during the coming years.

**Conflict of interest disclosure**

The author declares that he has no conflicts of interest.

**Data accessibility statement**

The estimates of NDVImax, climate layers and all ancillary data for selected pixels (coordinates, attribution to climate clusters) were uploaded to https://zenodo.org/XXX. All other data used in this study were from open-access sources, as described in the Materials and Methods section.

**Author contribution**

Single-authored paper.

**Acknowledgements**

Thanks are due to Brad Carlson, Arthur Bayle and Simon Gascoin for fruitful discussions during the preparation of this manuscript. This work received funding from the LIFE PASTORALP project (LIFE16 CCA/IT/000060) and from the Agence

Nationale de la Recherche (ANR) in the framework of the project TOP, Trajectories of agrO-Pastoral systems in mountains (ANR-20-CE32-0002). LECA is part of the Recherche Labex OSUG@2020 (IA-10-LABX-0056).





**Supporting Information**

The Supplement related to this article is available online at doi: XXX.




**Table 1. Structure and evaluation of competing linear mixed-effect models to characterize the variability of NDVImax for above-treeline ecosystems of the European Alps. Models were fitted using response and predictors for a total of 1053 cells (CELL) within 6 bioclimate clusters (CLU). MSE: maximum snow cover extent for the first 6 months; PRE: precipitation; WBA: atmospheric water balance (the difference between precipitation and potential evapotranspiration); T2M: air temperature; HWI: heat wave index. Meteorological variables correspond to total (PRE, WBA, HWI) or means (T2M) for the early summer months of June-July. The model in bold with the lowest AIC is the selected model. The Savitzky-Golay smoother was used as the NDVI curve fitting method.**

| Model formulation | | | | Model evaluation | | |
| Fixed effect | Random effect | Random structure | | AIC | Marginal variance (%) | Conditional variance (%) |
| | | intercept | slope | | | |
|---|---|---|---|---|---|---|
| MSE | CELL | fixed | random | 39993 | 4.3 | 4.3 |
| MSE | CLU / CELL | fixed | random | 39878 | 4.4 | 5.0 |
| MSE | CLU / CELL | random | random | 39886 | 4.4 | 5.0 |
| MSE + HWI | CLU / CELL | fixed | random | 39706 | 4.4 | 6.1 |
| MSE + T2M | CLU / CELL | fixed | random | 39084 | 4.2 | 8.9 |
| MSE + PRE | CLU / CELL | fixed | random | 38758 | 6.7 | 11.0 |
| MSE + WBA | CLU / CELL | fixed | random | 38792 | 6.2 | 10.7 |
| MSE + WBA + T2M | CLU / CELL | fixed | random | 37852 | 8.9 | 14.6 |
| MSE + WBA * T2M | CELL | fixed | random | 38618 | 8.7 | 12.5 |
| **MSE + WBA * T2M** | **CLU / CELL** | **fixed** | **random** | **37802** | **9.1** | **14.7** |
| MSE + WBA * T2M | CLU / CELL | random | random | 37816 | 9.1 | 14.3 |




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





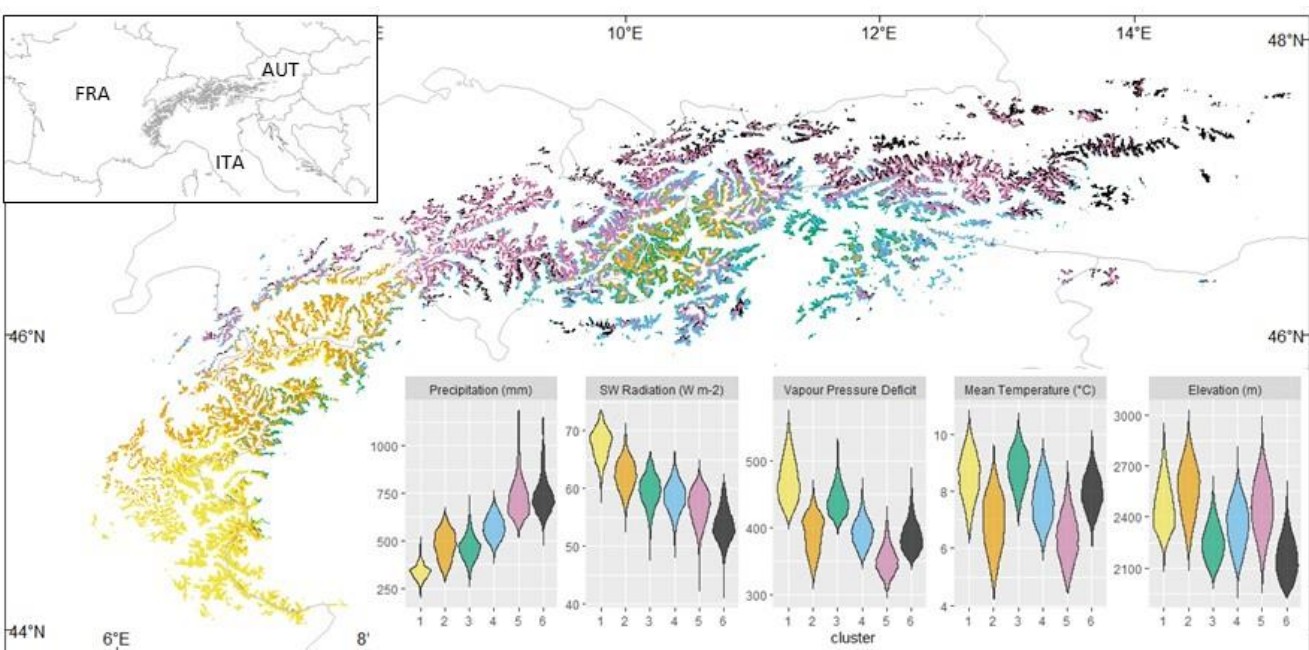

**Figure 1: Geographical distribution of above-treeline ecosystems in the European Alps and delineation of the six subregions based**
**on a clustering of summer climate averages for the reference period 1981-2010. The upper left panel shows the location of the study**
**area. Violin plots in the lower right panels quantify the distribution of bioclimate values between clusters. Climate data were**
**obtained from the high-resolution climatology CHELSA v2.1.**

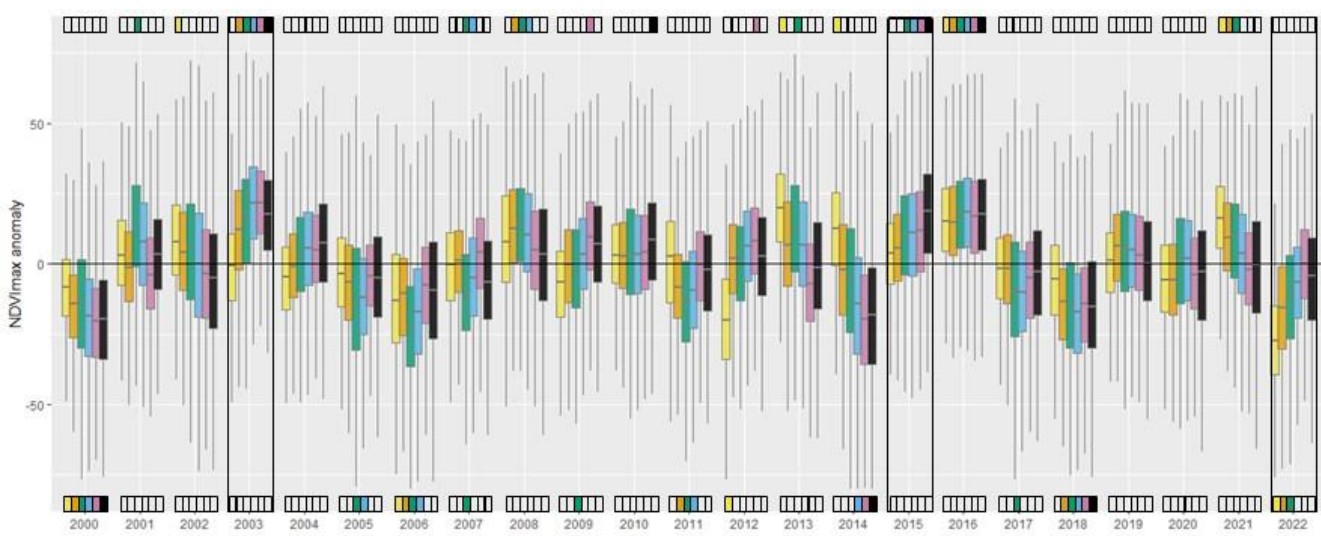

**Figure 2: Anomalies of NDVImax for the six clusters. Time series of NDVImax were detrended using a monotonic function and the**
**anomalies show the residual around the trend. Upper (lower) color chips indicate whether the collective, i.e. per cluster, anomaly**
**was significantly above (or below) 0, based on a counting of signs test for random samples of 10% of the pixels per cluster. To ease**
**reading, colored chips only indicate the most significant tests (P < 0.001). The three rectangles point to the three major heat waves**
**that have occurred in the Alps during the last two decades. Color legends of clusters are in Figure 1.**






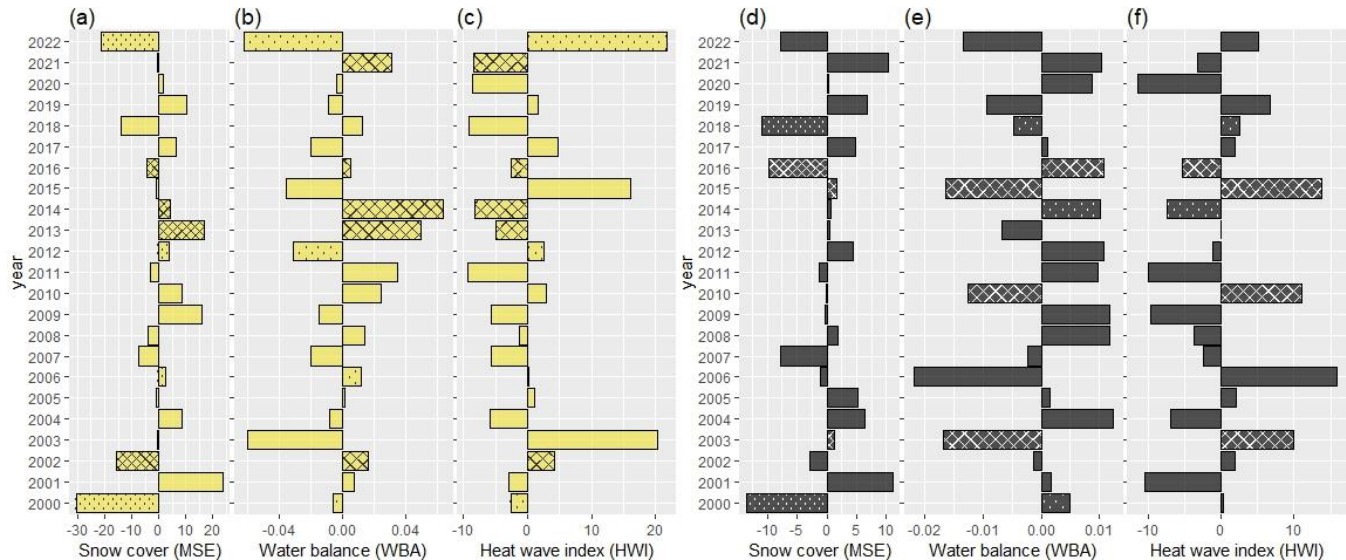

**Figure 3: Anomalies of snow cover duration (MSE) and early summer meteorological variables (WBA, HWI) for cluster 1 (a, b, c) and cluster 6 (d, e, f). Crosshatched (dotted) bars indicate significant positive (negative) NDVImax anomalies. Similar figures for the other clusters are given in Supplement Figure 2.**


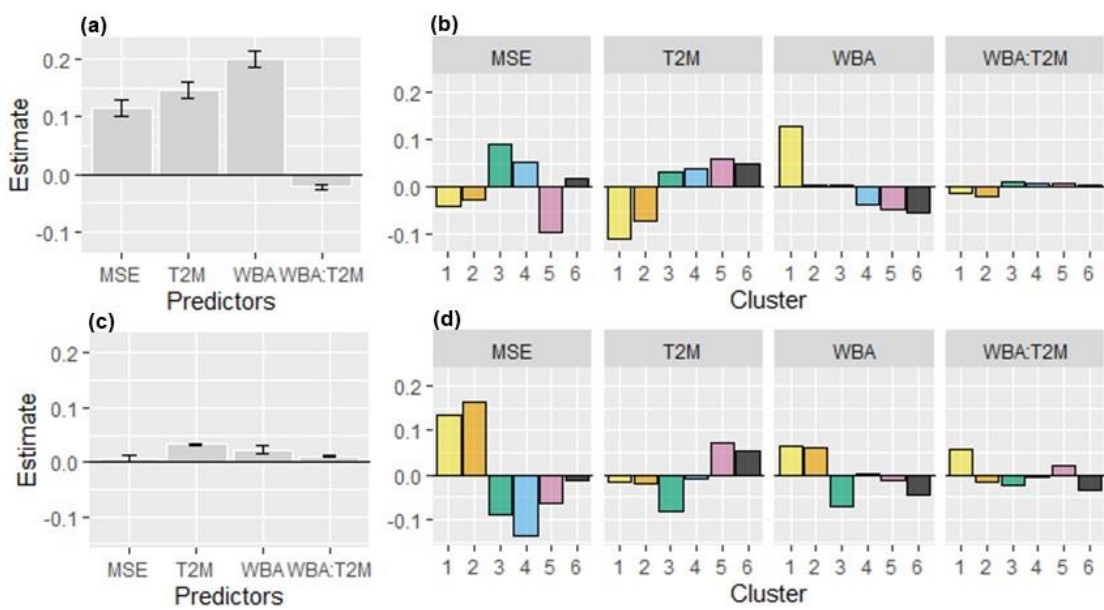

**Figure 4: Estimates of the fixed effects (gray bars) and of the cluster random effects (colored bars) for the predictors of NDVImax variability. Panels a, b show results for the full period (2000-2022), and panels c, d show the differences between the second and the first 13-year long period. Data correspond to the best linear mixed effect model outlined in Table 1. See Material and methods for**
**details.**