# Peer review of "Above-treeline ecosystems facing drought: lessons from the European 2022 summer heatwave"

_Biogeosciences, 2023_

## Author Response (AR1)

Dr. Philippe CHOLER

Univ. Grenoble Alpes, Univ. Savoie Mont Blanc, CNRS, LECA

F-38000, GRENOBLE

philippe.choler@univ-grenoble-alpes.fr

orcid : 0000-0002-9062-2721

Grenoble, 21 July 2023

Dear Editor,

I am very pleased to submit the revised version of the manuscript entitled "*Above-treeline ecosystems facing drought: lessons from the European 2022 summer heatwave*".

As detailed below, I have modified the text following the suggestions made by the two referees.

Yours sincerely,

Philippe Choler

**Response to referee 1.**

I thank the referee for his/her positive comments on the manuscript.

The study of Lamprecht et al. 2018, New Phytologist is now cited in the revised version of the introduction to provide a more balanced view on the reported trends of alpine species cover over the last decades.

All the typos have been corrected in the revised version.

**Response to referee 2.**

I thank the referee for his/her constructive comments on the manuscript.

The choice of MODIS NDVI was justified by the opportunity to track changes in vegetation greenness over the last 23 years, at a 250 m resolution based on a 1 to 2-day revisit period. This enabled (i) the comparison of several heat waves events (including that of 2003), (ii) the characterization of rapid changes in NDVI during the short growing season at high elevation and (iii) the implementation of a more robust statistical analysis of the drivers of NDVI response in a topographically heterogeneous landscape. To my knowledge, this would have not been possible with CGLS-derived vegetation indices, as this combination of a long timeframe, a short revisit period and a moderate spatial resolution has no equivalent.

To make it clear, I have provided a more elaborate rationale of the choice of MODIS in the revised version.

In a previous study on the long-term greening of the Alps (Choler & al, Global Change Biology, DOI: 10.1111/gcb.15820), we showed that NDVI-derived seasonal metrics were robust to the perturbation of the RED and NIR raw reflectance by up to ±5%. In addition, the different methods I used in this manuscript to reduce the noise of NDVI time series provided very similar results. I am therefore confident that MODIS NDVI was appropriate to support the main ecological conclusions of the manuscript.

As recommended, the discussion has been re-organized into subsections to better match the three addressed questions.

I have also modified the conclusion to emphasize that available high spatial resolution products (such as Sentinel-2) open avenues to characterize the response of specific alpine habitats to recent extreme events, including the 2022 heat wave.

All the typos and editorial comments have been accounted for in the revised version.

---

## Author Response (AR2)

Dr. Philippe CHOLER

Univ. Grenoble Alpes, Univ. Savoie Mont Blanc, CNRS, LECA

F-38000, GRENOBLE

philippe.choler@univ-grenoble-alpes.fr

orcid : 0000-0002-9062-2721

Grenoble, 14 August 2023

Dear Editor,

I am very pleased to submit the new revised version of the manuscript entitled "*Above-treeline ecosystems facing drought: lessons from the European 2022 summer heatwave*".

As recommended, I attempted to revise the figure legends, by including more links between them and by adding more context. I hope this has improved their readability.

Please note that the text has been edited by a B. Z. Carlson, a former PhD student of mine, who is a native from the U.S.A.

Yours sincerely,

Philippe Choler